# Child sexual abuse and its determinants among children in Addis Ababa Ethiopia: Systematic review and meta-analysis

**Birye Dessalegn Mekonnen**[1], **Sintayehu Simie Tsega**[2]*

1 Amhara Public Health Institute, Bahir Dar, Ethiopia, 2 Department of Medical Nursing, School of Nursing, College of Medicine and Health Science, University of Gondar, Gondar, Ethiopia

* simies952@gmail.com

## Abstract

Child sexual abuse is a significant public health concern and a breach of basic human rights affecting millions of children each year globally. It is typically not reported by victims, hence it remains usually concealed. Except for single studies with varying reports, there is no national studies conducted on child sexual abuse in Ethiopia. Therefore, this review determined the pooled magnitude and determinants of sexual abuse among children in Ethiopia. Potential articles were searched from PubMed, Science Direct, Scopus, and Web of science using relevant searching key terms. The Joanna Briggs Institute (JBI) critical appraisal checklist was used to evaluate the quality of all selected articles. Data were analyzed using STATA Version 14 software. Publication bias was checked using Egger's test and funnel plot. Cochran's chi-squared test and $I^2$ values were used to assess heterogeneity. A random-effects model was applied during meta-analysis. The pooled prevalence of sexual abuse among 5,979 children in Ethiopia was 41.15% (95% CI: 24.44, 57.86). Sex of children (OR: 2.14, 95%CI: 1.12, 4.06), smoking (OR: 4.48, 95%CI: 1.26, 76.79), khat chewing (OR: 3.68, 95%CI: 1.62, 21.93), and alcohol use (OR: 4.77, 95%CI: 2.22, 10.25) were the determinants of child sexual abuse. The main perpetrators of sexual abuse against children were neighbors, boy/girlfriends, family members, school teachers, and stranger person. Child sexual abuse commonly took place in the victim's or perpetrator's home, school, and neighbor's home. This review revealed that the magnitude of child sexual abuse in Ethiopia was relatively high and multiple factors determined the likelihood of sexual violence against children. Thus, policy-makers and concerned stakeholders should strengthen comprehensive sexual and reproductive health education to reduce the magnitude and consequences of child sexual abuse. Moreover, support with special attention should be given to children with mental illness and physical disabilities.

## Introduction

Child sexual abuse is the participation of a child (below the age of 18 years) in sexual activity that she or he is unable to give informed consent, does not fully understand, the child is not

**Data Availability Statement:** All data are in the manuscript and/or supporting information files.

**Funding:** The authors received no specific funding for this work.

**Competing interests:** The authors have declared that no competing interests exist.

developmentally prepared [1]. It includes a variety of activities such as sexual intercourse, attempted sexual intercourse, oral-genital contact, exposing children to pornography or adult sexual activity, fondling of genitals, and the use of the child for prostitution [2,3].

Child sexual abuse is a widespread and serious global public health problem and great violation of human rights that is now recognized as a public health priority [4]. Though sexual abuse affects people of all ages, genders, and sexual orientations, the majority of victims are women, children, and adolescents [5,6]. The global prevalence of child sexual abuse which has been estimated at 7.9% for males and 19.7% for females [7].

Child sexual abuse is a persistent form of child abuse that is often not reported, difficult to identify, and therefore often remains concealed [8]. It remains usually hidden in many developing countries as victims cannot report such assaults [9]. Factors that could contribute for underreporting include discrepancies in the definitions of sexual abuse, committed in complete privacy, fear of social stigma against survivors, and cultural and social norms [10,11].

Child sexual abuse is strongly associated with social determinants such as weak rule of law, poor governance, social, cultural and gender norms, low income, unemployment, limited educational opportunities, and gender inequality [12,13]. Likewise, factors such as social isolation, parental conflicts, lack of parental control, the absence of one or both parents, and family adversity have been associated to a higher risk of child sexual abuse [13–16].

Child sexual abuse has long-lasting effects and can be overwhelming for a child's relationships and social life [17,18]. Existing literature have revealed that childhood sexual abuse was associated psychosocial problems, psychiatric disorders, self-harm, and physical health problems such as sexually transmitted diseases including HIV, unwanted pregnancy, unsafe abortion, and obesity [19–23]. Similarly, systematic reviews have also indicated that sexual abuse in childhood was associated with psychosocial, psychiatric and health outcomes such as substance misuse, depression, post-traumatic stress disorder and anxiety over the life course [24–26].

In Ethiopian, there are no national pooled prevalence studies conducted on child sexual abuse except for individual studies with inconsistence reports. Additionally, identification of perpetrators, determinants, and consequences of child sexual abuse have not been well described. Thus, a systematic review and quantitative synthesis of findings from prevalence studies conducted on child sexual abuse is required. Therefore, this systematic review aimed to estimate the pooled prevalence and determinants of child sexual abuse in Ethiopia. Hereafter, recognizing the magnitude of child sexual abuse and its determinants helps to develop comprehensive strategies and interventions most fitted for children and inform policy-makers.

## Methods

The Preferred Reporting Items for Systematic Reviews and Meta-Analyses (PRISMA) checklist was used to prepare and report this systematic review and meta-analysis (S1 Checklist). The protocol for this systematic review was developed and registered on the International Prospective Register of Systematic Reviews (CRD42022362613).

### Eligibility criteria

Observational studies that report the prevalence and/ or determinants of child sexual abuse, studies that published and reported in the English language, both published and non-peer reviewed but publicly available studies conducted only in Ethiopia were considered. Articles that did not report the prevalence and/ or determinants of child sexual abuse, case reports, commentaries, case studies, and review articles were excluded.

## Information sources and search strategy

A systematic search of potentially relevant articles was performed from PubMed, Science Direct, Scopus, and Web of science until 28 August 2022. Literature search was conducted using the following keywords: "Magnitude", "Prevalence", "sexual abuse", "sexual violence", "sexual harassment", "rape", "sexual coercion", "sex offense", "child", "children", "under 18 years age", "below 18 years age", "associated factors", "determinants", "predictors" and "Ethiopia". The search was conducted using a variety of truncation, Boolean operators such as "OR" or "AND". Moreover, non-peer reviewed but publicly available articles were searched from Google.

## Study selection

Article screening process was performed using the EndNote X7.2.1 (Thomson Reuters, New York, USA) software citation manager. Two reviewers (BDM and SST) thoughtfully review the titles and abstracts of articles for relevance of the studies. Subsequently, full-text studies were retrieved and assessed to approve eligibility. The overall study selection processes were summarized using the PRISMA flow diagram.

## Data extraction and quality assessment

Data were extracted from included articles by two reviewers (BDM and SST) independently using the Joanna Briggs Institute (JBI) tool adapted for cross-sectional studies. The following data were collected from selected articles: first author name, study design, study setting, year of publication, sample size, response rate, number of participants, prevalence of child sexual abuse, perpetrators, determinants, consequences of child sexual abuse.

The methodological quality of all included articles was evaluated by two reviewers (BDM and SST) independently using the JBI critical appraisal checklist. The overall methodological quality of each article was based on the sum of points gained, which ranged from 0 to 10 points. Accordingly, all studies scored $\geq$ 60% of the JBI quality appraisal criteria and were included in the review. Any disagreements during data extraction and quality assessment were resolved through discussion with the third reviewer.

## Operational definitions

**Child**: is a person under the age of 18 years, unless the laws of a specific country set the legal age for adulthood earlier [27].

**Child sexual abuse**: is the involvement of a child in sexual activity such as intercourse, attempted intercourse that she or he is unable to fully understand and give informed consent [4]. In this review, child sexual abuse was considered if the primary studies reported any type or forms of sexual abuse such as completed or attempted sexual act.

## Data analysis

The results of included studies were described and summarized using figures, tables, and forest plots. Meta-analysis was executed using STATA 14. Heterogeneity effect sizes was assessed using the Q statistic and quantified by $I^2$ values. The existence of heterogeneity between included studies was presumed when $p < 0.1$ or $I^2 > 50\%$ [28]. A random-effects model was performed to execute the pooled prevalence of child sexual abuse as substantial heterogeneity was exhibited across included studies.

Publication bias was evaluated using Egger's test and inspection of asymmetry funnel plot. A p-value $\leq 0.05$ for Egger's test was indicative of presence of publication bias [29]. To assess

the presence effect of outliers, sensitivity analysis was performed [30]. All statistical analyses were considered statistically significant at a p-value of 0.05.

## Results

### Study selection

Initially, the search strategy produced 8,646 recorded literatures. A total of 2,785 records were removed because of duplication. Then, 5,861 articles were screened based on their titles and abstracts, which results in the removal of 5,820 articles. Consecutively, the remaining 41 full-text articles were independently assessed based on the inclusion criteria, which results in further exclusion of 33 articles. Full-text articles were excluded because of variation in study locations and population, not clearly report the outcome of interest, and some were reviews. Finally, eight studies were included in this systematic review (Fig 1).

### Study characteristics

In this review, a total of 5,979 children were included from an estimated 6,003 sample size. The sample size of the selected articles ranged from 327 [31] to 1,666 [32]. Regarding study design

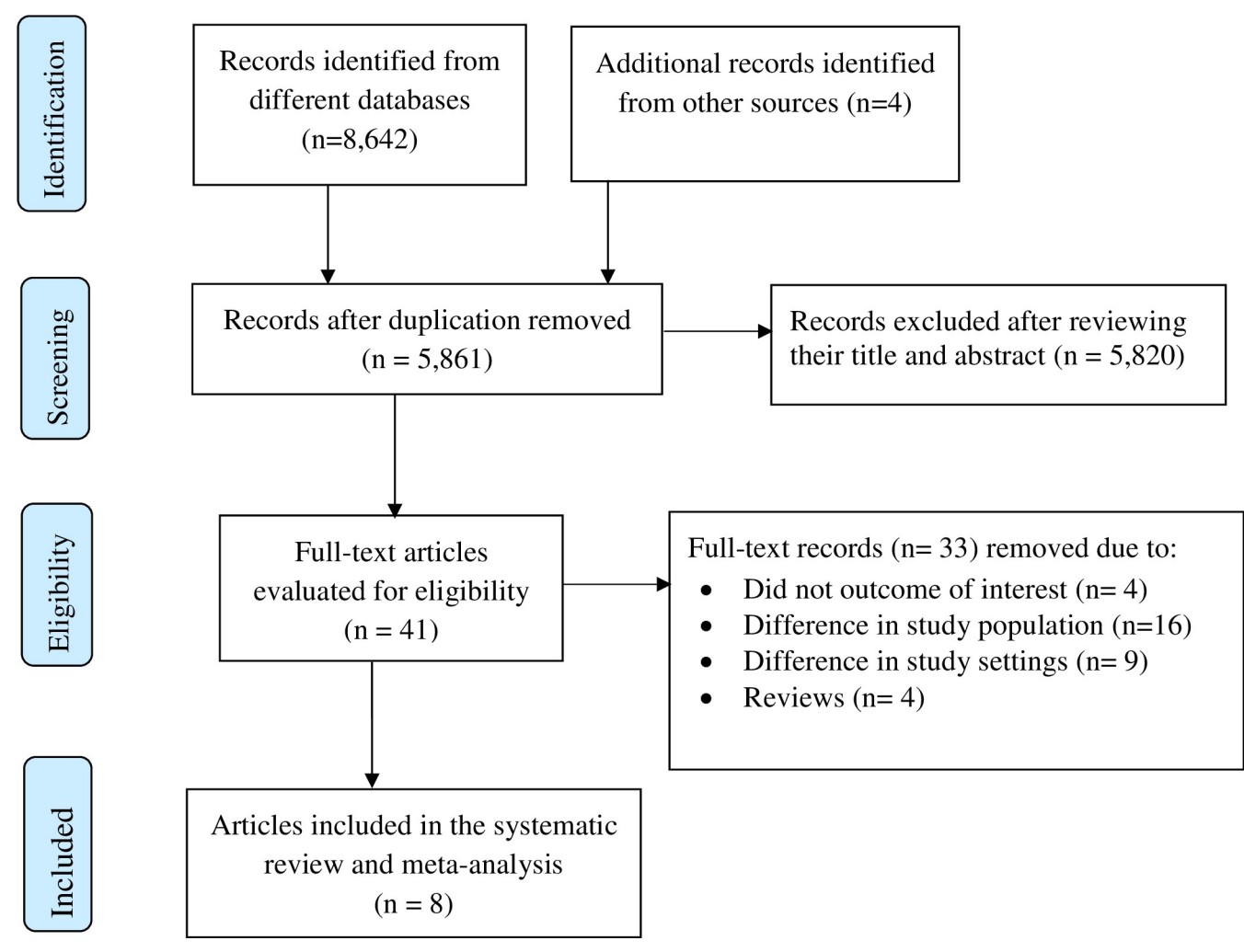

**Fig 1. PRISMA flow diagram for meta-analysis of child sexual abuse in Ethiopia.**

**Table 1. Summary of studies included in the meta-analysis of child sexual abuse in Ethiopia, 2022.**

| Author | Year | Study area | Study design | Sample size | Response rate | Number of participants | Outcome (event) | Prevalence (%) | Quality score |
|---|---|---|---|---|---|---|---|---|---|
| Abera et al [35] | 2021 | Dire Dawa | Cross-sectional quantitative | 794 | 98.8 | 785 | 384 | 48.9 | 9 |
| Alemayehu et al. [36] | 2022 | Addis Ababa | Cross-sectional quantitative | 422 | 100 | 422 | 180 | 42.7 | 9 |
| Assabu G et al [34] | 2019 | Addis Ababa | Cross-sectional quantitative | 1500 | 100 | 1500 | 1100 | 73.4 | 7 |
| Chemimdessa A et al [37] | 2014 | Addis Ababa | Mixed (Quantitative and Qualitative) | 422 | 96.4 | 407 | 160 | 39.3 | 6 |
| Habtamu D and A. Adamu [33] | 2013 | Addis Ababa | Mixed (Quantitative and Qualitative) | 422 | 100 | 422 | 55 | 13.1 | 8 |
| Jibril Jemal [32] | 2012 | Addis Ababa | Mixed (Quantitative and Qualitative) | 1666 | 100 | 1666 | 384 | 23.1 | 7 |
| Muluwork Tefera [31] | 2017 | Addis Ababa | Cross-sectional quantitative | 327 | 100 | 327 | 133 | 40.7 | 7 |
| Takele M et al [38] | 2020 | Addis Ababa | Cross-sectional quantitative | 450 | 100 | 450 | 217 | 48.2 | 8 |

of included studies, five studies were quantitative cross-sectional, and three studies employed a mixed (quantitative and qualitative) study design approach. Almost all (n = 7) of the studies were conducted in Addis Ababa. All studies included in this review were conducted from 2012 to 2021. The magnitude of child sexual abuse ranged from 13.1% [33] to 73.4% [34]. The result of the quality assessment indicated that one study was scored 6 points, three were scored 7 points, two were scored 8 points, and two were scored 9 points (Table 1).

## Sensitivity analysis and publication bias

The result of sensitivity analysis revealed that no individual studies significantly affect the pooled prevalence of child sexual abuse. Egger's rank test and Egger's test were performed to evaluate the presence of publication bias. Accordingly, the results of the tests indicated that publication bias was not detected, with the p-value of 0.956, and symmetry of the funnel plot (Fig 2).

## Prevalence of child sexual abuse

The pooled prevalence of sexual abuse among 5,979 children in Ethiopia was 41.15% (95% CI: 24.44, 57.86). In this meta-analysis, random-effects model was applied because of high heterogeneity ($I^2$ = 99.5%, p < 0.001) was detected across the included studies (Fig 3).

## Dealing with heterogeneity

Subgroup analysis and meta-regression were conducted to detect the potential sources of heterogeneity between studies. Furthermore, random-effects model was used during meta-analysis. The subgroup analysis was performed based on study design (cross sectional quantitative Vs mixed study). The result indicated that there was no significant difference in the level of heterogeneity across study designs used in the included studies (Fig 4).

A meta-regression analysis was done based on year of publication, study design and sample size. Accordingly, the results of the meta regression analysis showed all the variables had no significant effect on the pooled prevalence of child sexual abuse in Ethiopia (Table 2).

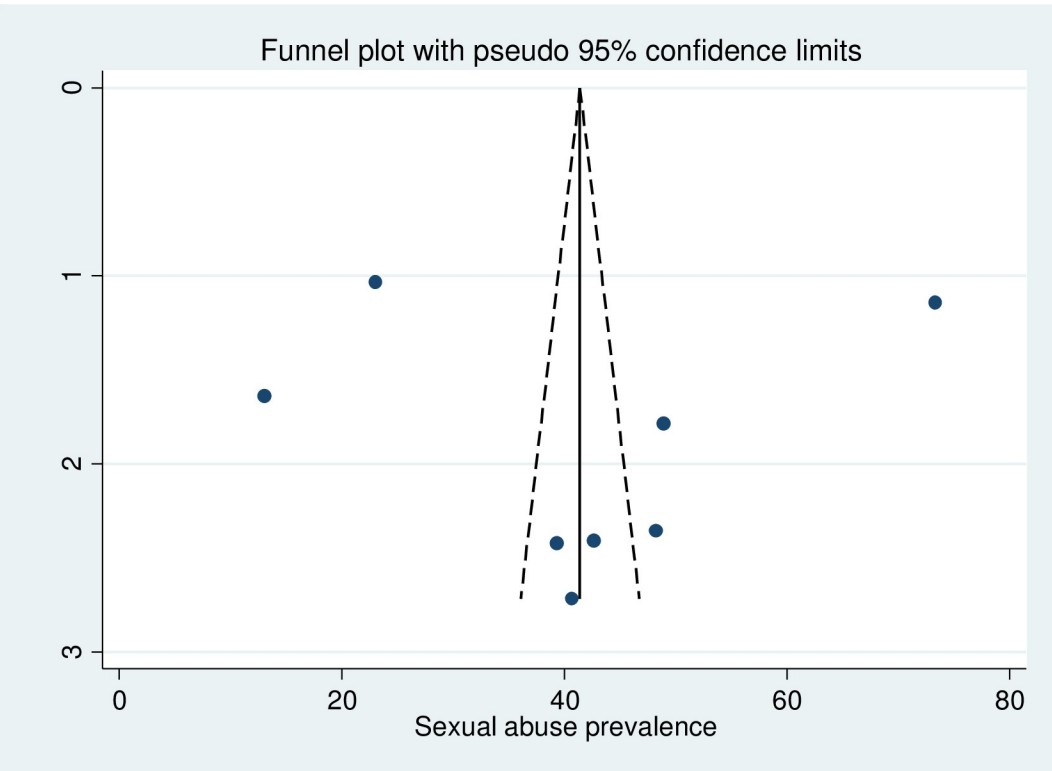

**Fig 2. Graphic representation of publication bias using funnel plots.**

## Perpetrators of child sexual abuse

The main perpetrators of sexual abuse against children were neighbors, with pooled prevalence of 34.3% (14.04, 54.59) followed by boy/ girlfriends at 15.9% (11.00, 20.80). Child sexual abuse commonly took place in the victim's or perpetrator's home, school, neighbor's home, Hotel, and in the public street. Table 3 shows the perpetrators of sexual abuse against children.

## Determinants of child sexual abuse

The factors associated with child sexual abuse that were consistently reported in more than one primary study were pooled quantitatively. Accordingly, the results of the meta-analysis showed that female children were about 2.14 times (OR: 2.14, 95%CI: 1.12, 4.06) more likely to experience sexual abuse. Furthermore, children who smoke cigarette (OR: 4.48, 95%CI: 1.26, 76.79), khat chewing (OR: 3.68, 95%CI: 1.62, 21.93), and alcohol use (OR: 4.77, 95%CI: 2.22, 10.25) were more likely to experience any type of sexual abuse (Table 4). There is likely an interaction between the variables of alcohol use, khat chewing, and gender, suggesting that boys are more likely to engage in these behaviours.

The determinants of child sexual abuse not included in the meta-analysis due to inconsistent categorization or being reported only in one primary study were systematically reviewed. Hence, children over the age of 14, and rural residents were more likely to experience sexual abuse [35,36,38]. Additionally, children who had mental illness and physical disability were more likely to be sexually abused [36]. Furthermore, children who did not have an open discussion with their parents on sexual and reproductive health, and living with their single parent were also more likely to experience sexual abuse [35,36].

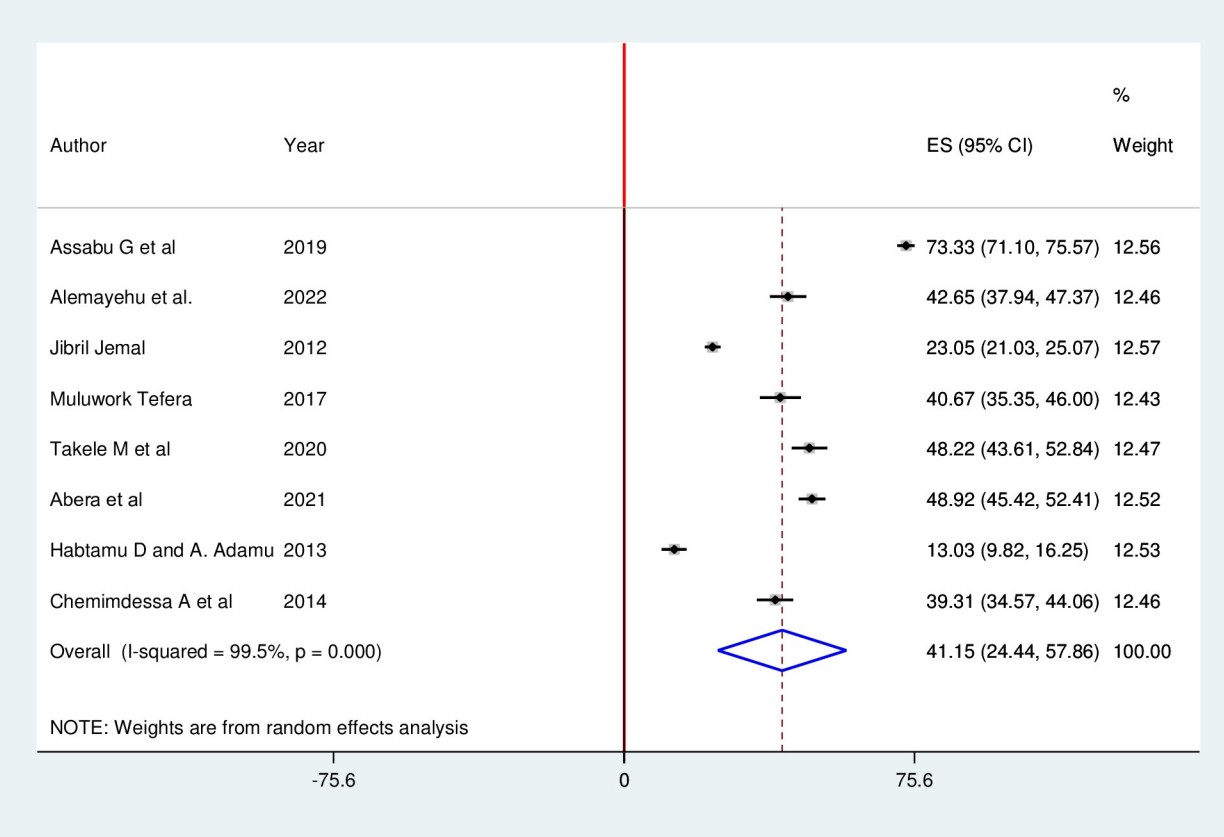

**Fig 3. Forest plot showing the pooled prevalence of child sexual abuse in Ethiopia, 2022.**

## Discussion

This meta-analysis showed that the pooled prevalence of child sexual abuse in Ethiopia was 41.15%. This finding was in line with the findings of a systematic review and meta-analysis that reported the prevalence of child sexual abuse as 32% [39]. This implies that a significant proportion of children experiencing sexual abuse in their lifetime, which requires the need of prioritizing intervention that reduce child sexual abuse.

Female children were more likely to experience sexual abuse than male children. The findings of a systematic review and meta-analysis also supported this finding, which indicated that female sex, was associated with increased risk of child sexual abuse [40]. This could be attributed by the effect of community perception towards gender roles results in the expectation that female be submissive to males [41,42]. Literature showed that masculinity, culture, social norms, and beliefs about gender roles substantially contribute to the high prevalence of sexual abuse among females [43,44].

Children who use alcohol were more likely to experience sexual abuse. This could be because drinking alcohol can change one's consciousness and ability to solve problems. In addition to increasing risk-taking, alcohol also causes people to become less conscious of and concerned about the effects of their actions [45]. Moreover, alcohol consumption decreases one's ability to decision making [46]. Similarly, the findings indicated that children who smoke cigarette, and chew khat were more likely to experience any type of sexual abuse. The stimulant effect of khat chewing exposes children to feelings of sexual activity such as watching pornographic movies, which make them to easily manipulated by perpetrators [47]. Evidence

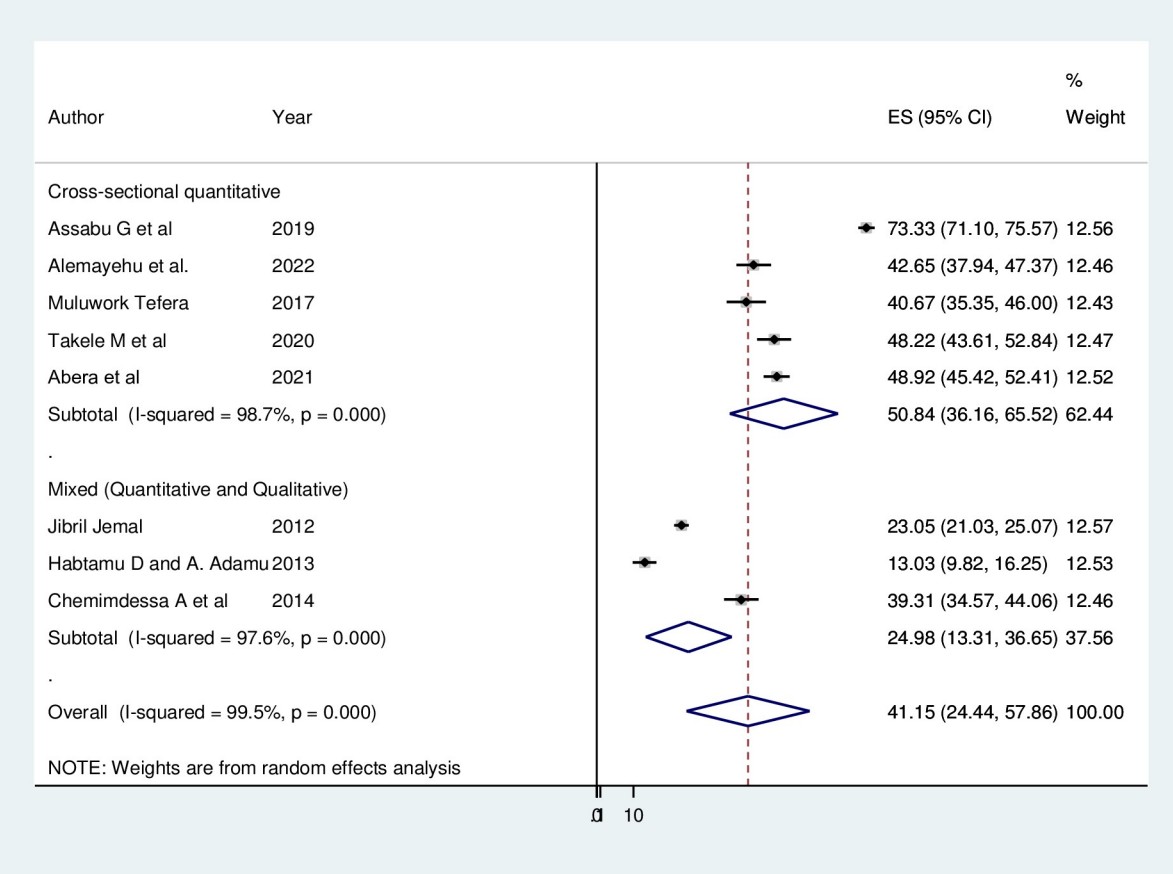

**Fig 4. Subgroup analysis of child sexual abuse in Ethiopia, 2022.**

indicated that perpetrators of sexual abuse usually use substances such as alcohol, chat, and cigarette as tool to take advantage of someone who is vulnerable and lacking capacity for decision making [48,49]. In this review, gender is identified as determinants of child sexual abuse. Hence, there is likely an interaction between the variables of alcohol use, khat chewing, and gender, suggesting that boys are more likely to engage in these behaviours [50,51].

Children who had a mental illness and physical disability were more likely to be sexually abused, which is supported by previous systematic reviews and meta-analysis [52]. This could be because of children living with disability are more vulnerable to emotional, sexual, and physical abuse and neglect than non-disabled children [53,54]. Similarly, children who lack open discussion on sexual issues with parents were more likely to experience sexual abuse, which is supported by a previous systematic reviews [55]. This could be due to less parental

**Table 2. Results of meta-regression for child sexual abuse in Ethiopia, 2022.**

| Variable | Std. Err. | Meta-regression Coefficient | (95%CI) | p-value |
|---|---|---|---|---|
| Publication year | 1.399 | 3.232 | (-0.192, 6.657) | 0.060 |
| Study design | | | | |
| Cross-sectional quantitative | 0.014 | 0.008 | (-0.025, 0.041) | 0.591 |
| Mixed (quantitative and qualitative) | 5.771 | -0.241 | (-14.363, 13.881) | 0.968 |
| Sample size | 0.014 | 0.008 | ( -0.025, 0.041) | 0.589 |

**Table 3. Perpetrators of child sexual abuse in Ethiopia, 2022.**

| Perpetrators of child sexual abuse | Studies | Estimates (95% CI) | Pooled prevalence % (95% CI) | Test of heterogeneity | |
|---|---|---|---|---|---|
| | | | | I² (%) | P |
| Neighbors | Assabu G et al. | 28.8 (26.51, 31.09) | 34.3 (14.04, 54.59) | 99.19 | 0.000 |
| | Muluwork Tefera | 54.6 (49.20, 59.99) | | | |
| | Abera et al. | 19.9 (17.11, 22.69) | | | |
| Boy/girlfriends | Assabu G et al. | 17.5 (15.58, 19.42) | 15.9 (11.00, 20.80) | 93.22 | 0.000 |
| | Alemayehu et al. | 13.3 (10.06, 16.54) | | | |
| | Muluwork Tefera | 8.6 (5.56, 11.64) | | | |
| | Abera et al. | 23.8 (20.82, 26.78) | | | |
| | Chemimdessa A et al. | 16.2 (12.62, 19.78) | | | |
| Stranger persons | Assabu G et al. | 4.8 (3.72, 5.88) | 12.7 (1.88, 27.28) | 99.45 | 0.000 |
| | Muluwork Tefera | 27.9 (23.04, 32.76) | | | |
| | Abera et al. | 5.9 (4.25, 7.55) | | | |
| Family members | Assabu G et al. | 23.6 (21.45, 25.75) | 11.8 (1.79, 21.82) | 99.04 | 0.000 |
| | Muluwork Tefera | 5.8 (3.27, 8.33) | | | |
| | Abera et al. | 16.7 (14.09, 19.31) | | | |
| | Chemimdessa A et al. | 1.2 (0.14, 2.26) | | | |
| School teachers | Assabu G et al. | 4.5 (3.45, 5.55) | 9.09 (1.19, 19.38) | 99.25 | 0.000 |
| | Muluwork Tefera | 3.4 (1.44, 5.36) | | | |
| | Abera et al. | 25.0 (21.97, 28.03) | | | |
| | Chemimdessa A et al. | 3.7 (1.87, 5.53) | | | |

involvement in children's sexual and reproductive health behaviors which leads to missing opportunities to acquire experiences and life skills for the prevention of sexual abuse. Hence, there should be a parent-child open discussion on sexual issues without considering it as shameful and taboo.

The neighbors, family members, school teachers, boy/girlfriends, and stranger person were identified as the most common perpetrators of sexual abuse against children. This is supported by an existing literature which indicated that children were most commonly subjected to sexual abuse by family members, neighbors, relatives, and other persons known to the victim child [56,57]. While the primary studies reported boy/girlfriends (romantic friends) as perpetrators of child sexual abuse without specifying the more common gender, it was found that boys are more likely to engage in sexual offense behaviors [58]. This review also indicated that

**Table 4. Determinants of child sexual abuse in Ethiopia, 2022.**

| Variables | Studies | Odds ratio (95% CI) | Pooled odds ratio % (95% CI) | Test of heterogeneity | |
|---|---|---|---|---|---|
| | | | | I² | P |
| Sex | Alemayehu et al. | 1.9 (0.79, 4.72) | 2.14 (1.12, 4.06) | 0.00 | 0.75 |
| | Takele M et al. | 2.4 (0.94, 6.0) | | | |
| Alcohol | Alemayehu et al. | 2.2 (1.36, 3.53) | 4.77 (2.22, 10.25) | 89.84 | 0.001 |
| | Takele M et al. | 5.97 (3.84, 9.28) | | | |
| | Abera et al. | 8.02 (5.63, 11.43) | | | |
| Chat | Alemayehu et al. | 3.6 (1.46, 28.69) | 3.68 (1.62, 21.93) | 0.00 | 0.15 |
| | Takele M et al. | 3.8 (1.11, 31.68) | | | |
| Cigarette | Alemayehu et al. | 5.9 (1.05, 42.12) | 4.48 (1.26, 76.79) | 0.00 | 0.30 |
| | Takele M et al. | 3.8 (1.11, 36.02) | | | |

child sexual abuse commonly took place in the victim's or perpetrator's home, school, neighbor's home, hotel, and in the public street. Literature also indicated that perpetrators of sexual abuse are more likely to commit the attack in the victim's or perpetrator's home [59,60]. This implies that sexual offenders might consciously plan to commit sexual abuse.

The findings of this systematic review indicated that victims of child sexual abuse experienced psychological consequences such as feeling lonely, attempt to commit suicide, fear of males, and verbal and physical aggressiveness. This finding is supported by previous systematic reviews [24,55,61–64]. Similarly, unwanted pregnancies, underwent an abortion, pain during urination, and vaginal discharge were sexual and reproductive health consequences reported by victims of child sexual abuse, which is supported by previous findings [24,61,63,65].

## Limitations of the study

The following limitations should be considered when reading and interpreting the findings of this study: Firstly, though meta-regression and subgroup analyses were done, the sources of high heterogeneity across included studies was not identified. Secondly, the generalizability of the review might not be in full confidence since the included studies were not from all the regions of Ethiopia, almost all the studies were from Addis Ababa. Subsequently, regional differences were not discussed because of the small number of studies from a limited geographic area. Thirdly, the pooled odds ratio for some factors were not examined because of inconsistent categorization. Fourthly, discussion of the findings comparing with existing evidence couldn't be done due to lack of comparable studies in the study country and worldwide. Nevertheless, this systematic review provided the first pooled magnitude of child sexual abuse in Ethiopia, to the best of the authors knowledge.

## Implications of the study

This study revealed that nearly half of children experienced sexual abuse by different perpetrators. This finding infers the need of strengthening policy or regulations interventions specific to child sexual abuse by in view of them as they are disadvantaged group. The finding also implies the need of better performing for awareness creation on the concept of child sexual abuse for the society. Besides, the finding could be attributed to the role of understanding the burden of child sexual abuse for the likelihood and success of prevention and control interventions. Literature also indicated that understanding the burden of child sexual abuse would help for the likelihood and effectiveness of prevention and control strategies.

The current review provided vibrant evidence to notify policy-makers, and other concerned stakeholders to prevent and control child sexual abuse. Some determinants were identified that are associated with increased experiences of child sexual abuse. While the use of substances was identified as a determinant of child sexual abuse, we did not obtain clear information on legislation regarding substance or drug use among children in Ethiopia. Thus, policy-makers should develop and enforce comprehensive regulations to address and reduce substance use among children. In addition, prioritizing the determinants and the prevention of child sexual abuse should be started sooner rather than later. Furthermore, health information and education provision programs are necessary to empower the society. Moreover, special support should be given for the victims. Perpetrators of child sexual abuse were identified as individuals typically involved in such offenses against children. Therefore, enforcing appropriate policies and legislation such as encouraging children and parents to report such acts, public awareness creation campaigns, prompt punishment of the perpetrators are crucial to end child sexual abuse. However, it is important to consider that many survivors tend not to report abuse which may affect the prevalence estimates. Research indicated that survivors of sexual

abuse did not report the abuse due to a lack of awareness of how and to whom to report, a desire to protect the family, fear that the abuser will retaliate, and financial dependence on the perpetrator [66,67].

## Conclusion

This review revealed that the magnitude of child sexual abuse in Ethiopia was relatively high and multiple factors determined likelihood of sexual violence against children. Therefore, policy-makers, and other concerned stakeholders should design and implement interventions that could empower the community in their struggle toward the prevention and elimination of child sexual abuse, create awareness on the burden and consequences of child sexual abuse, prompt punishment of the sex offenders, and control and monitor the implementation of policies and legislation. Moreover, support with special attention should be given to female children and those who use alcohol, chat, and cigarette.

## Supporting information

**S1 Checklist. Preferred Reporting Items for Systematic Reviews and Meta-Analyses (PRISMA) checklist.**
(DOC)

## Author Contributions

**Conceptualization:** Birye Dessalegn Mekonnen, Sintayehu Simie Tsega.

**Data curation:** Birye Dessalegn Mekonnen, Sintayehu Simie Tsega.

**Formal analysis:** Birye Dessalegn Mekonnen, Sintayehu Simie Tsega.

**Funding acquisition:** Birye Dessalegn Mekonnen, Sintayehu Simie Tsega.

**Investigation:** Birye Dessalegn Mekonnen, Sintayehu Simie Tsega.

**Methodology:** Birye Dessalegn Mekonnen, Sintayehu Simie Tsega.

**Project administration:** Birye Dessalegn Mekonnen, Sintayehu Simie Tsega.

**Resources:** Birye Dessalegn Mekonnen, Sintayehu Simie Tsega.

**Software:** Birye Dessalegn Mekonnen, Sintayehu Simie Tsega.

**Supervision:** Birye Dessalegn Mekonnen, Sintayehu Simie Tsega.

**Validation:** Birye Dessalegn Mekonnen, Sintayehu Simie Tsega.

**Visualization:** Birye Dessalegn Mekonnen, Sintayehu Simie Tsega.

**Writing – original draft:** Birye Dessalegn Mekonnen, Sintayehu Simie Tsega.

**Writing – review & editing:** Birye Dessalegn Mekonnen, Sintayehu Simie Tsega.

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
