## [Decision Letter · Decision Letter 0]

23 Aug 2023

PGPH-D-23-00743

Child Sexual Abuse and Its determinants Among Children in Ethiopia: Systematic review and Meta-analysis

Dear Dr. Simie Tsega,

Thank you for submitting your manuscript to PLOS Global Public Health. After careful consideration, we feel that it has merit but does not fully meet PLOS Global Public Health’s publication criteria as it currently stands. Therefore, we invite you to submit a revised version of the manuscript that addresses the points raised during the review process.

We have secured two reviewers for your paper - please address each comment thoughtfully and thoroughly in your response and in your revisions. Many thanks.

We look forward to receiving your revised manuscript.

Kind regards,

Julia Robinson

Executive Editor

Journal Requirements:

1. We noticed you have some minor occurrence of overlapping text with the following previous publication(s), which needs to be addressed:

- https://reproductive-health-journal.biomedcentral.com/articles/10.1186/s12978-022-01470-2

- https://www.dovepress.com/magnitude-of-child-sexual-abuse-and-its-associated-factors-among-child-peer-reviewed-fulltext-article-AHMT

In your revision ensure you cite all your sources (including your own works), and quote or rephrase any duplicated text outside the methods section. Further consideration is dependent on these concerns being addressed.

Additional Editor Comments (if provided):

Reviewers' comments:

Reviewer's Responses to Questions

**Comments to the Author**

1. Does this manuscript meet PLOS Global Public Health’s publication criteria? Is the manuscript technically sound, and do the data support the conclusions? The manuscript must describe methodologically and ethically rigorous research with conclusions that are appropriately drawn based on the data presented.

Reviewer #1: Yes

Reviewer #2: Yes

2. Has the statistical analysis been performed appropriately and rigorously?

Reviewer #1: Yes

Reviewer #2: Yes

3. Have the authors made all data underlying the findings in their manuscript fully available (please refer to the Data Availability Statement at the start of the manuscript PDF file)?

Reviewer #1: Yes

Reviewer #2: Yes

4. Is the manuscript presented in an intelligible fashion and written in standard English?

Reviewer #1: No

Reviewer #2: No

5. Review Comments to the Author

Reviewer #1: Child Sexual Abuse and Its determinants Among Children in Ethiopia: Systematic review and Meta-analysis

This is quite an educative study which has not been researched much in Africa. There are few corrections necessary for this paper to be fit for publication.

The 5th paragraph from the introduction, this sentence is incomplete “Child sexual abuse has long-lasting effects and overwhelming on a child” Overwhelming what?

In the study characteristics, what was the researchers quality assessment based upon?

The information in the result section are inadequate. I would expect this researcher to provide more detailed information on what was found and some of the reasons to the variations in the prevalence of child sexual abuse across different and even some same location of Ethiopia. More so, I expected that the gravity of the determinants of this act would be discussed in details especially to reveal what the commonest determinants were in that order. same with the results on the perpetrators and consequences. The researcher should provide details on how many studies identified what factors or perpetrators or consequences, this will be very relevant to this study.

I would also suggest the review of the repetition of same words at the beginning of every paragraph. This should be reviewed.

Reviewer #2: Review of "Child Sexual Abuse and Its determinants Among Children in Ethiopia: Systematic review and Meta-analysis"

The authors present a systematic review and meta-analysis estimating the pooled prevalence and determinants of child sexual abuse in Ethiopia based on 8 studies with 5,979 participants. They find a high pooled prevalence of 41.15%, indicating child sexual abuse is a major public health issue in Ethiopia. The review has several strengths, including its systematic methods, meta-analytic approach to pool prevalence, and comprehensive data extraction about determinants and consequences. The topic is highly important given the immense burden of child sexual abuse globally.

However, some major revisions are needed to strengthen the manuscript:

Major Revisions:

1. There was high statistical heterogeneity (I2= 99.5%) in the meta-analysis, but sources were not explored through subgroup analysis or meta-regression based on study-level factors like location, sample, etc. Examining potential sources will clarify if the heterogeneity impacts the overall pooled estimate.

2. The authors identified several determinants from their narrative synthesis, but should consider meta-analyzing these if enough studies examined the same factors consistently. This will provide pooled effect sizes rather than just a descriptive summary.

3. The limitations around generalizability need to be expanded, given the small number of studies from a limited geographic area. Discussion of regional differences would strengthen this point.

Minor Revisions:

1. Some previous studies require fuller citations in the discussion.

2. Some minor edits to improve writing clarity and flow would further enhance readability.

3. Biases inherent in child sexual abuse research should be briefly mentioned regarding how they may impact prevalence estimates.

Overall Recommendation:

I recommend major revision to address the comments above. Doing so will significantly strengthen the manuscript. The topic is highly important, the review fairly rigorous, and the major revisions feasible. Addressing the heterogeneity, meta-analyzing determinants, and expanding the discussion will make this a strong candidate for publication. I look forward to reviewing the revised manuscript, which holds good promise for contributing meaningful evidence synthesizing the current knowledge on child sexual abuse in Ethiopia.

6. PLOS authors have the option to publish the peer review history of their article (what does this mean?). If published, this will include your full peer review and any attached files.

**Do you want your identity to be public for this peer review?** For information about this choice, including consent withdrawal, please see our Privacy Policy.

Reviewer #1: **Yes: **Queen Esther Adeyemo

Reviewer #2: **Yes: **Alok Atreya

---

## [Decision Letter · Decision Letter 1]

4 Jan 2024

PGPH-D-23-00743R1

Child Sexual Abuse and Its determinants Among Children in Ethiopia: Systematic review and Meta-analysis

Dear Mr Tsega,

Thank you for submitting your manuscript to PLOS Global Public Health. After careful consideration, we feel that it has merit but does not fully meet PLOS Global Public Health’s publication criteria as it currently stands. Therefore, we invite you to submit a revised version of the manuscript that addresses the points raised during the review process.

We look forward to receiving your revised manuscript.

Kind regards,

Lana Clara Chikhungu, PhD

Academic Editor

Journal Requirements:

2. We noticed that you used "unpublished" in the manuscript. We do not allow these references, as the PLOS data access policy requires that all data be either published with the manuscript or made available in a publicly accessible database. Please amend the supplementary material to include the referenced data or remove the references.

Additional Editor Comments (if provided):

There is a need to ensure that the title indicates that the studies reviewed are largely from Addis Ababa and not the whole of Ethiopia.

Can you indicate how the response rate was calculated and indicate whether the response rate mentioned above includes quantitative and mixed methods studies?

Determinants of child sexual abuse

There is likely an interaction between the variables, alcohol use,  khat chewing and gender.  I take that it is the boys who are more likely to engage in these behaviours.  This needs to be discussed.

Reviewers' comments:

Reviewer's Responses to Questions

**Comments to the Author**

Please check comments from Reviewer 1. 

Reviewer #1: (No Response)

Reviewer #2: All comments have been addressed

2. Does this manuscript meet PLOS Global Public Health’s publication criteria? Is the manuscript technically sound, and do the data support the conclusions? The manuscript must describe methodologically and ethically rigorous research with conclusions that are appropriately drawn based on the data presented.

Reviewer #1: Yes

Reviewer #2: Yes

3. Has the statistical analysis been performed appropriately and rigorously?

Reviewer #1: Yes

Reviewer #2: Yes

4. Have the authors made all data underlying the findings in their manuscript fully available (please refer to the Data Availability Statement at the start of the manuscript PDF file)?

Reviewer #1: Yes

Reviewer #2: Yes

5. Is the manuscript presented in an intelligible fashion and written in standard English?

Reviewer #1: Yes

Reviewer #2: Yes

6. Review Comments to the Author

Reviewer #1: Review: Child Sexual Abuse and Its determinants Among Children in Ethiopia: Systematic review and Meta-analysis

This systematic review and meta-analysis on child sexual abuse among children in Ethiopia make a significant and commendable contribution to our understanding of a critical issue. By synthesizing existing studies, the research not only sheds light on the prevalence of child sexual abuse but also identifies determinants, offering valuable insights for policymakers and practitioners. This study serves as a crucial step forward in addressing a sensitive topic, paving the way for targeted interventions and fostering a safer environment for children in Ethiopia.

However, there are few clarifications that needs to be done on this paper, which are listed below;

1. Perpetrators of child sexual abuse

In your report of findings here, I don’t think you meant to mention family members and school teachers, as the main perpetrators by the pooled prevalence were neighbors and boys/girls. By the way what does the authors mean by boys/girls? I think a better way to report this will be to separate the sex in order to reveal what gender perpetrates more.

2. Determinants of child sexual abuse

According to your review, only gender and abuse of substances were reported to be the determinants of child sexual abuse, this seems to be inadequate, peradventure they are the most prevalent determinants, it is essential to mention other determinant according to the reviewed articles. There are a lot of factors that could predispose to child sexual abuse, how about children who do not use the reported substances?

3. Discussion: In the paragraph on the perpetrators, the second sentence; “This is supported by an existing literature which indicated that children were most commonly perpetuated by family members, neighbors, relatives, and other persons known to the victim child”...perpetuated by what? do you mean to mention sexually perpetuated?

4. Limitations and Implication of study, should be discussed under a separate sub-heading.

5. In the last paragraph before the conclusion, the second sentence “. Perpetrators of child sexual abuse were identified that are usually committing sexual abuse against children” should be rewritten for clear understanding.

6. The authors did not discuss any implication of this study on the determinants of child sexual abuse. What are the legislation concerning the use of these substances among children in Ethiopia? What can recommendations based on the findings from this study can help to limit the gravity of the substance use and thus child sexual abuse?

7. Other general comments:

Check your page numbering, it wasn’t consistent, also multiple line spacing were used, be consistent on either single-spaced or double-spaced.

Reviewer #2: General comments:

The authors have carefully addressed previous reviewer comments and made substantial improvements to the manuscript overall. The introduction frames the importance of child sexual abuse and rationale for the systematic review and meta-analysis well. The methods and analysis appear rigorous, with clear reporting of the search, inclusion criteria, quality assessment, data analysis, and risk of bias procedures. Limitations regarding generalization have been expanded appropriately. I have only minor suggestions to further strengthen the paper:

Abstract:

- The abstract clearly summarizes key aspects of the study. Consider adding a line on implications/recommendations from the findings to conclude the abstract.

Introduction:

- Well written with good flow. No specific suggestions.

Methods:

- The methods are reported in adequate detail. No further suggestions.

Results:

- The presentation of results is much improved from the initial submission, especially related to perpetrators and consequences of abuse. Consider adding frequencies of how many studies examined each perpetrator/consequence to contextualize percentages.

- For determinants meta-analysis, state reasons if you were unable to pool any identified factors (e.g, inconsistent reporting across studies).

Discussion:

- The discussion covers the central findings and situates them well compared to past literature. To further highlight implications, consider adding another paragraph on tangible next steps/recommendations stemming from this review.

- References are now appropriately incorporated. Just double check journal name abbreviations match reference guidelines.

Minor edits:

- Carefully proofread for any leftover grammatical issues, especially subject-verb agreement.

Overall, this systematic review addresses an important public health issue and provides meaningful insight into the magnitude and factors associated with child sexual abuse in the Ethiopian context. With only minor revisions to augment discussion/conclusions, this paper will represent a strong contribution to the growing literature in this area and have potential to impact policy and practice.

7. PLOS authors have the option to publish the peer review history of their article (what does this mean?). If published, this will include your full peer review and any attached files.

**Do you want your identity to be public for this peer review?** For information about this choice, including consent withdrawal, please see our Privacy Policy.

Reviewer #1: **Yes: **Queen Esther Adeyemo

Reviewer #2: **Yes: **Alok Atreya

---

## [Editor Report · Decision Letter 2]

30 Jan 2024

PGPH-D-23-00743R2

Child Sexual Abuse and Its determinants Among Children in Ethiopia: Systematic review and Meta-analysis

Dear Dr. Tsega,

Thank you for submitting your manuscript to PLOS Global Public Health. After careful consideration, we feel that it has merit but does not fully meet PLOS Global Public Health’s publication criteria as it currently stands. Therefore, we invite you to submit a revised version of the manuscript that addresses the points raised during the review process.

EDITOR:

The revision has addressed all the key comments.  However there are some minor comments that should be taken into account.  Since almost all papers are from Addis Ababa and not a good representation of Ethiopia as a whole you should consider changing the title, to something like:  **Child sexual abuse and its determinants among children in Addis Ababa Ethiopia: Systematic review and Meta-analysis**

**Other revisions are:**

**Line 178: Check sentence construction**

**Lines 229 to 231: **There is likely an interaction between the variables, alcohol use,  khat chewing and gender.  I consider that it is the boys who are more likely to engage in these behaviours.  This needs to be discussed.

I agree with the decision to leave out the response rate

We look forward to receiving your revised manuscript.

Kind regards,

Lana Clara Chikhungu, PhD

Academic Editor
---

## [Editor Report · Decision Letter 3]

29 Feb 2024

Child sexual abuse and its determinants among children in Addis Ababa Ethiopia: Systematic review and Meta-analysis

PGPH-D-23-00743R3

Dear Dr Tsega

We are pleased to inform you that your manuscript 'Child sexual abuse and its determinants among children in Addis Ababa Ethiopia: Systematic review and Meta-analysis' has been provisionally accepted for publication in PLOS Global Public Health.

Best regards,

Lana Clara Chikhungu, PhD

Academic Editor

Reviewer Comments (if any, and for reference):

Thank you for changing the title of the manuscript to indicate Addis Ababa Ethiopia, however can you also reflect this change in the abstract.